# Self-Adaptively Learning to Demoiré from Focused and Defocused Image Pairs

Lin Liu[1,2]   Shanxin Yuan[2]*   Jianzhuang Liu[2]   Liping Bao[1]   Gregory Slabaugh[2]   Qi Tian[3]

[1]EEIS Department, University of Science and Technology of China
[2]Noah's Ark Lab, Huawei Technologies        [3]Huawei Cloud BU
{ll0825,baoliping}mail.ustc.edu.cn {shanxin.yuan, liu.jianzhuang, gregory.slabaugh, tian.qi1}@huawei.com

## Abstract

Moiré artifacts are common in digital photography, resulting from the interference between high-frequency scene content and the color filter array of the camera. Existing deep learning-based demoiréing methods trained on large scale datasets are limited in handling various complex moiré patterns, and mainly focus on demoiréing of photos taken of digital displays. Moreover, obtaining moiré-free ground-truth in natural scenes is difficult but needed for training. In this paper, we propose a self-adaptive learning method for demoiréing a high-frequency image, with the help of an additional defocused moiré-free blur image. Given an image degraded with moiré artifacts and a moiré-free blur image, our network predicts a moiré-free clean image and a blur kernel with a self-adaptive strategy that does not require an explicit training stage, instead performing test-time adaptation. Our model has two sub-networks and works iteratively. During each iteration, one sub-network takes the moiré image as input, removing moiré patterns and restoring image details, and the other sub-network estimates the blur kernel from the blur image. The two sub-networks are jointly optimized. Extensive experiments demonstrate that our method outperforms state-of-the-art methods and can produce high-quality demoiréd results. It can generalize well to the task of removing moiré artifacts caused by display screens. In addition, we build a new moiré dataset, including images with screen and texture moiré artifacts. As far as we know, this is the first dataset with real texture moiré patterns.

## 1   Introduction

Image demoiréing is the task of removing moiré patterns from images, taken by digital cameras from screens or from natural images with high-frequency patterns. Moiré artifacts are caused by the interference between the color filter array (CFA) of a camera and high-frequency repetitive signals, which can result from an LCD screen's subpixel layout or a natural scene's high-frequency repetitive patterns (e.g., textures on clothes). Image demoiréing is challenging as the moiré patterns vary in shape, color, and frequency. Existing deep learning based demoiréing models [39, 20, 12, 50, 24] rely heavily on training with large amounts of annotated clean and moiré image pairs in order to obtain good performance. However, the models are still limited in handling various complex moiré patterns. Moreover, these models are restricted to perform demoiréing of images captured from screens, having difficulty in removing moiré artifacts from natural images.

Lack of high-quality training data also limits the performance of supervised methods. There are two public datasets for screen image demoiréing, which are TIP2018 dataset [39] and LCDMoire dataset [46, 47]. TIP2018 is a real dataset with slight misalignment between each image pair and LCDMoire

---

is a synthetic dataset. Because both datasets were developed for screen image demoiréing, they are unsuitable to train a model to remove moiré patterns from images of high-frequency textures.

To reduce moiré artifacts, some digital-camera manufacturers design special hardware, including special CFAs (Fuji's X-Trans, Sigma SD Quattro) and variable low-pass filters (Sony RX1RM2). Special CFAs and variable low-pass filters require special hardware design and thus cannot be widely used on smartphones.

In this paper, we propose a self-adaptive learning method for image demoiréing. Our method removes moiré patterns from a moiré image with the help of a moiré-free blur image. We design a defocusing method to model the low-pass filter to obtain the blur image without moiré patterns. We use it as an additional input (defocused moiré-free image) to help remove moiré patterns from the moiré image (focused), and treat it as a joint filtering problem. Our method can be easily applied to any digital camera. During the focusing process, a defocused blur image can be stored and combined with the focused image to perform image demoiréing. Deep image prior [17] shows that the structure of a generator network can capture a great deal of low-level image statistics without any training. In our model, we use a 3-layer fully connected sub-network to generate a blur kernel, and adopt a U-Net-like encoder/decoder architecture to perform image demoiréing. Neither network is learned in an explicit training stage; but rather they are learned at test-time through an iterative, self-adaptive optimization.

In summary, our main contributions are:

1. We propose a self-adaptive learning method for image demoiréing, which uses an additional input (defocused moiré-free image) to help remove moiré patterns from the focused moiré image.

2. We create a new dataset with pairs of focused moiré and defocused moiré-free images, containing both screen moiré images and high-frequency texture moiré images[2].

3. Quantitative and qualitative experimental results on both public and our datasets show that our model outperforms state-of-the-art methods.

4. Our method, without a training stage, can be easily applied to any digital camera or smartphone.

## 2 Related Work

In this section, we review the most relevant work, including image demoiréing, joint filtering, self-adaptive learning, and blind deblurring.

**Image Demoiréing.** There are two common scenarios: *screen image* demoiréing and *texture image* demoiréing. Screen image demoiréing focuses on removing moiré patterns from photos taken from screens, where moiré patterns are mainly caused by the interference between the screen's subpixel layout and the camera's color filter array. Texture image demoiréing deals with moiré patterns that are produced by photographing high-frequency scene content (e.g., fabric and long-distance buildings), which interferes with the CFA. Early work [37, 33, 36] on screen image demoiréing focus on certain specific moiré patterns (striped, dotted or monotonous moiré patterns). Recently, some deep learning models [39, 20, 12, 24, 50] cast screen demoiréing as an image restoration problem and can handle more types of moiré patterns. Liu *et al.* [20] built a coarse-to-fine convolutional neural network to remove moiré patterns from photos taken from screens. Sun *et al.* [39] proposed a multi-resolution convolutional neural network for demoiréing and released an associated dataset. He *et al.* [12] labeled the data in [39] with three attribute labels of moiré patterns, which is beneficial to learn diverse patterns. These methods are all supervised and need training with a large-scale dataset. Moreover, after training, they cannot generalize well to texture image demoiréing.

Unlike screen image demoiréing, removing moiré patterns in texture images is more challenging as moiré patterns appear only at the high-frequency areas and are always mixed with the underlying textures. Recently, some researchers [44, 21, 45] have attempted to handle texture image demoiréing. Yang *et al.* [44] and Liu *et al.* [21] tried to remove moiré artifacts using low-rank and sparse matrix decomposition. Moiré patterns are also common artifacts from the image signal processing pipeline

in a camera, especially from image demosaicing [7, 23]. Gharbi *et al.* [7] proposed to alleviate the moiré artifacts by fine-tuning their demosaicing model on a moiré-prone dataset, which was collected by measuring the frequency change from the ground-truth image and the demosaiced image. Our network uses an additional input (defocused moiré-free image) to help remove moiré patterns from the moiré image and does not need training.

**Joint Filtering.** Joint filtering has been applied to many low-level vision tasks [31, 8], with the aim of leveraging the guidance image as a prior and transferring the structural details to the target image. It has good ability in handling images from different domains. Many applications have been tried including depth/RGB image restoration [8], flash/no-flash image denoising [31], texture removal [10, 49], etc. Local joint filtering methods [40, 4, 13, 49] make use of a locally linear model to explore the relationship among neighboring pixels. Representative methods include bilateral filtering [40, 4] and guided filtering [13]. But these methods usually introduce erroneous structures into the target image because they only explore the local structures of the guidance image. Global joint filtering methods [9, 5, 42] optimize a global objective function. Different hand-crafted priors were proposed to enforce the target image and guidance image to have similar structures. But the hand-crafted priors may not reflect inherent structural details in the target image. Recently, some deep learning based joint filtering algorithms [19, 41, 28] have been proposed and shown better results. Pan *et al.* [28] presented spatially variant linear representation coefficients, which are determined by both the guidance image and the input image, to decide whether the structural details should be transferred to the output image.

Our method can also be viewed as a joint filtering method, taking two inputs (a focused moiré image and a defocused blur image). The defocused blur image provides important structural information to guide the demoiréing network. The moiré patterns, especially low-frequency patterns, can be treated as a new structure overlaid on the structures of the moiré-free images. With the defocused image as a guide, the demoiréing network can enhance the original structural information and suppress the moiré structures. Unlike these deep learning based joint filtering methods, ours does not need training and produces a moiré-free image only from a defocused and focused image pair.

**Self-Adaptive Learning.** Self-adaptive learning has been used in some specific low-level vision tasks such as super-resolution [35, 14, 3], deblurring [25, 1, 32], inpainting [48] and dehazing [2]. They exploit the internal recurrence of information in an image without training. Recently, some researchers proposed some frameworks which can deal with multiple low-level vision tasks. Lempitsky *et al.* [17] showed that the structure of the deep image prior (DIP) neural network is very good to capture the low-level statistics of a single natural image. Gandelsman *et al.* [6] proposed a double-DIP framework for decomposing a single image into two layers. However, these two networks are unable to generate good results in the demoiréing problem, partly because moiré patterns are widely distributed in the spatial and frequency domains.

**Blind Deblurring.** Blind image deblurring is a very challenging problem because it needs to estimate both the blur kernel and the clean image from a blur image. Blind image deblurring methods can be divided into optimization-based and deep-learning based. Optimization-based methods use different priors for modeling clean images, such as gradient-based prior [30, 51], patch-based prior [25, 38] and dark channel prior [43, 29]. For modeling accurate blur kernels, gradient sparsity prior [18, 29] and spectral prior [22] are usually adopted. In most cases, it is a blind deblurring problem to deblur a defocused image from an uncalibrated camera. Unlike these methods, we can obtain a clear image with moiré patterns through focusing. This image contains rich details and benefits restoring a clean moiré-free image. We also design a generative network to estimate the blur kernel.

## 3 The Proposed Method

We first introduce the formulation of the problem, then design the network structure and finally present our algorithm.

### 3.1 Problem Formulation

When the image is defocused, image blur is spatially invariant in the same depth and the blur image $B$ can be formulated as,

$$B = K \otimes C + N_\sigma, \tag{1}$$

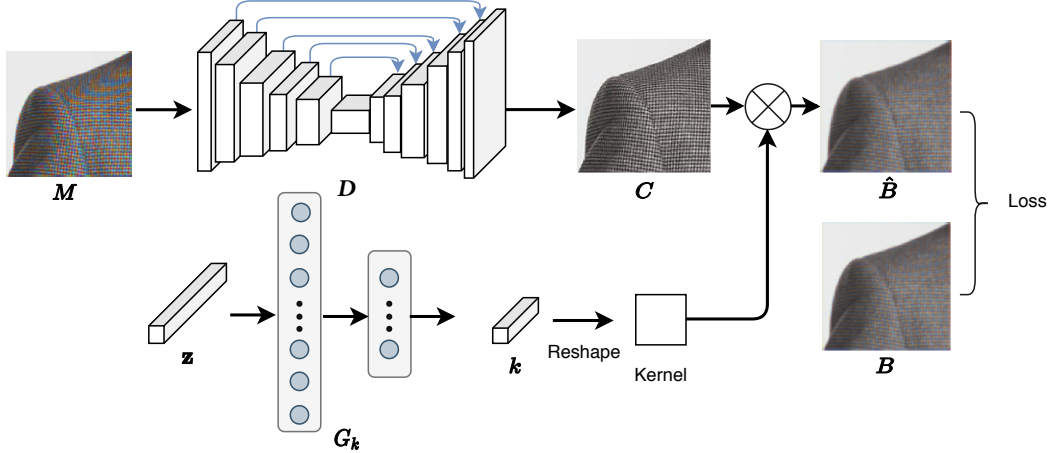

Figure 1: Illustration of our model. The generative network $G_k$ is utilized to obtain a prior of the blur kernel. The network $D$ can output a moiré-free image after multiple iterations. The whole network is self-learned using only the focused image $M$ with moiré patterns and the defocused image $B$ without moiré patterns.

where $C$ is the underlying clean image, $K$ is the blur kernel, $N_\sigma$ is the additive Gaussian noise with noise level $\sigma$, and $\otimes$ denotes 2D convolution. In most cases where the camera is not calibrated, we need to estimate both $K$ and $C$ from a blur image $B$, which is an ill-posed problem.

When the image is focused, moiré artifacts may appear in high-frequency areas. To remove the moiré patterns, we suppose an image demoiréing network $D$ can obtain the clean image such that

$$C = D\left(M\right), \tag{2}$$

where $M$ is a focused image contaminated with moiré patterns. Putting Eqns. 1 and 2 together, we have

$$B = K \otimes D\left(M\right) + N_\sigma. \tag{3}$$

Thus, the blur image $B$ is obtained by first removing the moiré patterns from $M$ and then convolving with the blur kernel $K$. In this combined formulation, Eqn 3, the demoir'eing network $D$ can be learned without using the underlying clean image $C$ as the ground truth.

Inspired by the DIP framework [17], we propose to use a generative network $G_k$ to capture the blur kernel $K$ as a prior. Finally the demoiréing problem is formulated as

$$\min_{D, G_k} \left\| G_k\left(\mathbf{z}\right) \otimes D\left(M\right) - B \right\|^2, \tag{4}$$

where $\mathbf{z}$ is a fixed vector and is sampled from the uniform distribution [0,1]. $G_k\left(\mathbf{z}\right)$ is the estimated blur kernel using the generator $G_k$. However, only optimizing Eqn. 4 cannot guarantee that $D$ will give good demoiréing results. Following [18, 29], we add the following constraints to the blur kernel,

$$\left(G_k\left(\mathbf{z}\right)\right)_i \geq 0, \ \forall i, \tag{5}$$

$$\sum_i \left(G_k\left(\mathbf{z}\right)\right)_i = 1, \tag{6}$$

where $\left(G_k\left(\cdot\right)\right)_i$ denotes the $i$-th element in the blur kernel. Note that in Section 4 we will describe how to obtain $B$.

## 3.2   Network Structure

Fig. 1 shows the structure of our model. The input of $D$ is a focused image with moiré patterns. $D$ is a U-Net-like network where its first 5 layers of the encoder are connected via skip-connections to the 5 layers of the decoder. A convolutional output layer with the sigmoid function is used to generate the moiré-free image $C$. U-Net-like structures have been shown to work well in many low-level computer

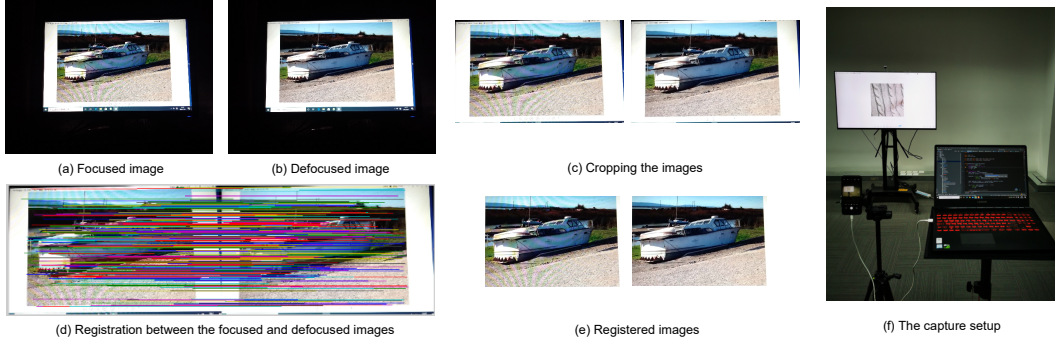

(a) Focused image          (b) Defocused image              (c) Cropping the images

(d) Registration between the focused and defocused images          (e) Registered images          (f) The capture setup

Figure 2: Illustration of image acquisition.

vision tasks [15, 39, 17]. For the network $G_k$, a blur kernel usually contains much less information than an image, and can be well estimated by a simpler generative network. Thus, we adopt a 3-layer fully-connected network (FCN) to serve as $G_k$. It takes a 200-dimentional vector (noise) $z$ as the input. The hidden layer and the output layer have 1,000 nodes and $K^2$ nodes, respectively, and the blur kernel size is $K \times K$. A softmax layer is applied to the output layer of $G_k$ to ensure the constraints in Eqns. 5 and 6.

### 3.3 Optimization Algorithm

The optimization process of Eqn. 4 can be viewed as a self-adaptive learning method. With only the defocused moiré-free image $B$ and the focused moiré image $\hat{M}$, the networks $D$ and $G_k$ iteratively find better weights, making $D$ produce clear images $\hat{C}$ with fewer and fewer moiré patterns. The parameters of $D$ and $G_k$ are simultaneously updated by back-propagation. Based on extensive experiments, we find that the joint optimization of $D$ and $G_k$ is better than the alternating optimization[3] of them (see the ablation study in Section 5.1).

## 4 A New Dataset

We create a new dataset to evaluate our method quantitatively and qualitatively, as there is no public dataset available for the specific setting in this paper.

**Synthetic Data:** For the screen moiré patterns, the data are sampled from TIP2018 dataset [39], from which we randomly choose 130 image pairs (moiré images and moiré-free images). The moiré images serve as the focused moiré images and the moiré-free images serve as the ground truth. To synthesize a defocused moiré-free image, we apply a Gaussian smoothing kernel (with $\sigma$ from 0.8 to 1.6) and an additive Gaussian noise (with noise level from 0 to 0.2) to the moiré-free image using Eqn. 1. We assume a simplified case where the whole image has the same blurriness. We call this subset *SynScreenMoire*.

For texture moiré patterns, we collect 30 high-quality images (with dense and regular textures) from the Internet and treat them as ground truth. The method in [44] is adopted to synthesize the corresponding moiré images. Finally, we use the same method for the screen moiré images to synthesize the defocused images from the ground truth. This subset is called *SynTextureMoire*.

**Real Data:** We build a real dataset with 100 pairs, each with a focused moiré image and a defocused moiré-free image. It includes 50 pairs where the moiré patterns are caused by the interference between the camera CFAs and the screen pixel layouts (this subset is called *RealScreenMoire*), and another 50 pairs where the moiré patterns are caused by the interference between the camera CFAs and the high-frequency textures of the images (this subset is called *RealTextureMoire*). As shown in Fig. 2, to capture a pair of images, we design an image acquisition pipeline, which mainly consists of two steps: image capture and image alignment.

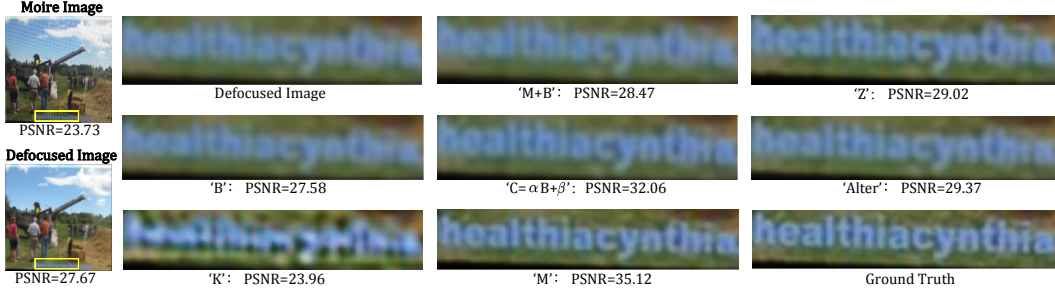

Figure 3: One example of the ablation study.

| Network | M+B | Z | B | $C = \alpha B + \beta$ | Alter | K | M |
|---|---|---|---|---|---|---|---|
| PSNR/SSIM | 27.43/0.852 | 28.42/0.869 | 27.17/0.842 | 29.57/0.846 | 27.31/0.855 | 26.10/0.734 | **31.77**/**0.926** |

Table 1: Ablation study on *SynScreenMoire*.

**(1) Image Capture.** Each image is displayed at the centre of a computer screen (Fig. 2(a) and (b)) and the background color of the screen is set to white for better alignment. To produce a wide variety of moiré patterns, we use three types of smartphones (OPPO R9, HONOR 9 and HUAWEI P30 PRO). For different image pairs, we randomly change the distance and angle between the camera and the computer screen. The cameras are placed on a tripod. It is worth noting that when capturing texture moiré images, the camera is farther away from the screen than when acquiring screen moiré images, to avoid screen moiré patterns. By adjusting the distance between the camera and the screen, when the displayed high-frequency image textures (perceived by the camera) have frequencies similar to the camera's CFA, texture moiré patterns appear. Since the frequencies of the image textures are much lower than the frequencies of the computer screen's subpixel layout, screen moiré patterns are minimized. In order to avoid camera shaking when changing the focus and defocus settings, we use a laptop to remotely control the zooming and shooting of the mobile phones. The capture process is shown in Fig. 2(f). We use a screen with 4k resolution to display the images when we make the *RealTextureMoire* subset. Using different phone or camera models as our capture devices ensures that the moiré patterns are across different optical sensors, while the diversity of display screens for *RealScreenMoire* exhibits the difference in screen resolution[4]. **(2) Image Alignment.** As the focal length increases, the objects in the image will appear larger. We need to register each defocused and focused image pair. With the help of a white background, we first binarize the captured image to find its area and then crop it. We then align an image pair using the homography [11].

## 5 Experiments

In this section, we show an ablation study and comparisons with state-of-the-art methods. Our algorithm is implemented in Pytorch. The experiments are conducted on a NVIDIA RTX 2080Ti GPU. In the kernel generation network, **z** is sampled from the uniform distribution in [0,1] with a fixed random seed 0. The initial learning rate is set to 0.01 and reduced by a half for every 500 iterations. The algorithm runs for 3000 iterations for each image pair.

### 5.1 Ablation Study

The *SynScreenMoire* subset is used to verify that the network $D$ can extract sufficient information from the focused image. The results are shown in Table 1, where 'M+B' means that the input to the network is the concatenation of the focused and the defocused images; 'B' denotes the input is only the defocused image; 'Z' means the input is a 2D noise image (sampled from the uniform distribution in [0,1]) of the same size as the focused image; 'M' stands for our original input (focused moiré image). Comparing 'M' with 'M+B' shows that adding the defocused moiré-free image as an additional input decreases the performance. We speculate that the whole network is confused by

| | Method | DMCNN [39] | CFNet [20] | MopNet [12] | DIP [17] | GF [13] | DJF [19] | MSJF [34] | FDNet |
|---|---|---|---|---|---|---|---|---|---|
| | S or U? | S | S | S | U | U | U | U | U |
| *SynScreenMoire* | PSNR/SSIM | 26.15/0.869 | 25.62/0.820 | 26.45/0.856 | 22.57/0.757 | 27.23/0.808 | 31.06/0.898 | 22.82/0.785 | **31.77/0.926** |
| *SynTextureMoire* | PSNR/SSIM | 22.79/0.714 | 22.08/0.702 | 23.44/0.789 | 22.51/0.720 | 21.99/0.525 | 22.40/0.752 | 24.70/0.687 | **25.98/0.794** |

Table 2: Quantitative comparison on *SynScreenMoire* and *SynTextureMoire*. S and U in the second row refer to Supervised and Unsupervised, respectively. The best results are highlighted in bold.

| | Method | MopNet [12] | DIP [17] | DoubleDIP [6] | GF [13] | DJF [19] | MSJF [34] | SVLRM [28] | FDNet |
|---|---|---|---|---|---|---|---|---|---|
| | S or U? | S | U | U | U | S | U | S | U |
| *RealScreenMoire* | NIQE/BRISQUE | 5.57/30.53 | 5.35/33.89 | 5.69/45.92 | 6.42/45.91 | 5.75/30.96 | 5.34/**29.34** | 9.42/31.04 | **5.11**/29.87 |
| *RealTextureMoire* | NIQE/BRISQUE | 17.85/42.81 | 18.73/42.53 | 13.64/48.44 | 11.99/51.84 | 21.65/42.13 | 11.67/45.21 | 12.63/42.74 | **11.22**/**41.77** |

Table 3: Quantitative comparison of image demoiréing on *RealScreenMoire* and *RealTextureMoire*. The best results are highlighted in bold and the second best are underlined.

this blur image in the input. This also verifies the necessity of learning a blur kernel. Comparing 'M' with 'Z' and 'B' shows that the network $D$ does extract useful information from the focused moiré image. Adding the moire image provides a good local minimum for the network to converge to. In fact, although the moiré image is corrupted by moire patterns, it still retains many high-frequency details. The U-NET can retain these details in the iteration for the reconstruction of a clean image from the blur image that lacks high-frequency details. As shown in Fig. 3, the result from 'M' is better than those from 'Z' and 'B'. Another baseline is by treating the final result as a spatially variant linear representation [28] of the defocused moiré-free image ($C = \alpha B + \beta$). The result shows that our end-to-end training ('M') is more powerful than predicting the parameters ($\alpha$ and $\beta$) of the image. We also compare our joint optimization method with the alternating optimization method ('Alter' in Table 1), which optimizes $D$ and $G_k$ alternately (see the supplementary material for more details). Moreover, we test the necessity of using a network (FCN) to generate the blur kernel by replacing the FCN with a learnable kernel, noted as 'K' in Table 1. The result shows that the PSNR/SSIM of 'K' is smaller than that of 'M'. Adding the FCN to learn the blur kernel can make the image satisfy the total variation prior and smooth the noise. In addition, we also tried the other architecture of $D$, the encoder-decoder and ResNet. Their PSNRs are 0.11dB and 1.05dB, respectively lower than U-NET. Many other image restoration methods have shown that U-NET has advantages over the two structures.

## 5.2 Comparison with State-of-the-Art

We compare our method with state-of-the-art demoiréing methods (DMCNN [39], CFNet [20] and MopNet [12]), joint filtering methods (GF [13], MSJF [34], DJF [19] and SVLRM [28]) and unsupervised image restoration methods (DIP [17] and DoubleDIP [6]). We also compare with some state-of-the-art blind deblurring methods to show that our model obtains useful information from the focused moiré images. For all blind deblurring methods and our method, the blur kernel and noise level are unknown. In what follows, we term our model FDNet since it uses both the Focused and Defocused image pair for demoiréing.

**Evaluation on SynScreenMoire and SynTextureMoire.** On the synthetic data with ground truth, we can use the Peak Signal-to-Noise Ratio (PSNR) and the Structural Similarity Index Measure (SSIM) to compare the restored images. The supervised methods, DMCNN, CFNet, MopNet and DJF are trained on TIP2018 for testing on *SynScreenMoire*, and trained on MITMoire for testing on *SynTextureMoire*. As shown in Table 2, FDNet outperforms all the state-of-the-art methods. We also perform experiments on *SynTextureMoire* to compare with the state-of-the-art blind deblurring methods to show FDNet extracts useful information from the focused moiré images. For DCP [29] and DeblurGANv2 [16], the PSNRs/SSIMs are 28.53/0.875 and 28.58/0.864, respectively, which are smaller than FDNet's 31.77/0.926.

**Evaluation on RealScreenMoire and RealTextureMoire.** On the real data without ground truth, we evaluate all generated images using no-reference quality metrics, which estimate absolute image quality scores. For objective quality measurement, we use the Blind/Referenceless Image Spatial Quality Evaluator (BRISQUE) [26] and Naturalness Image Quality Evaluator (NIQE) [27]. BRISQUE extracts the point-wise statistics of local normalized luminance signals and measures image naturalness. NIQE is based on the construction of a quality-aware collection of statistical features based

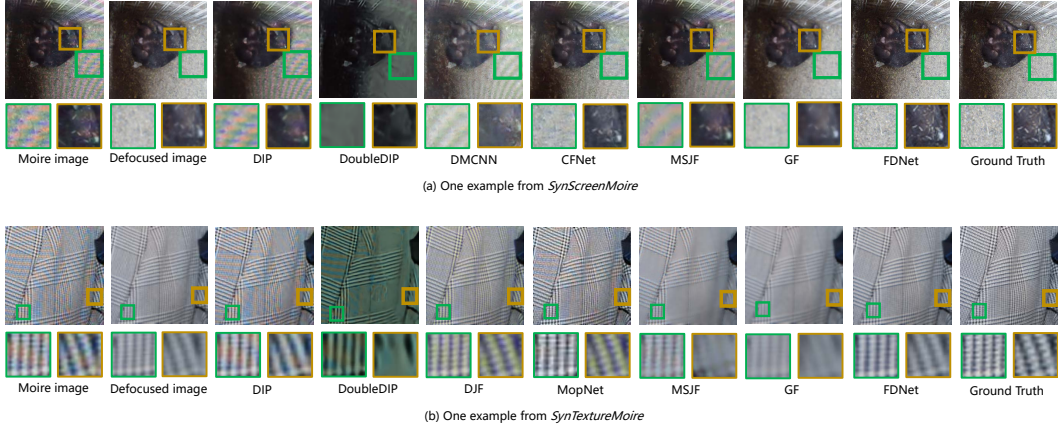

Moire image　Defocused image　DIP　DoubleDIP　DMCNN　CFNet　MSJF　GF　FDNet　Ground Truth

(a) One example from *SynScreenMoire*

Moire image　Defocused image　DIP　DoubleDIP　DJF　MopNet　MSJF　GF　FDNet　Ground Truth

(b) One example from *SynTextureMoire*

Figure 4: Visual comparison among our FDNet and other models, evaluated on images from *Syn-ScreenMoire* and *SynTextureMoire*.

on a simple space domain natural scene statistic model. Note that a smaller NIQE or BRISQUE score means an image with better quality (lower is better). An off-the-shelf trained algorithm (MAT-LAB2018b) is used to obtain the NIQE and BRISQUE scores. To compare with the existing methods, we randomly choose 25 images and 38 images from our real dataset *RealScreenMoire* and *RealTextureMoire*, respectively. The supervised methods MopNet, DJF and SVLRM are trained on TIP2018 for testing on *RealScreenMoire*, and trained on MITMoire for testing on *RealTextureMoire*. As shown in Table 3, our method overall outperforms both the supervised and unsupervised methods.

**Qualitative Results.** As shown in Fig. 4, the result of DIP has obvious moiré artifacts left, and DoubleDIP has a global color shift from the original input and over-smoothed details. These two deep image prior methods cannot effectively remove moiré patterns, perhaps because they struggle to learn the low-frequency characteristics and the color diversity of moiré patterns. Moire patterns and noise are different; the former are prevalent more in low and mid-frequencies. DIP relies on the spectral bias of the CNN to learn lower frequencies first. So before DIP learns the high-frequency details of the image, moire patterns have appeared in the results of DIP. The demoiréing only methods (DMCNN, CFNet and MopNet) cannot effectively remove the moiré patterns. In addition, the joint filtering methods (GF and MSJF) tend to blur the high-frequency regions and cannot remove the moiré patterns well with the guidance of the blur image. In contrast, our method FDNet eliminates the moiré patterns more effectively, benefiting from the accurate prediction of the blur kernel. In addition, FDNet retains the original textures in the images with moiré patterns removed instead of over-smoothing the high-frequency regions. More results are provided in the supplementary material.

## 5.3 Practical Applications of our Method

Our method has the potential of being applied to smartphones without modifying the hardware. In the typical capture mode, the camera usually is embedded with an auto-focus algorithm, which can be modified to save an additional defocused image. Unlike variable hardware low-pass filters in some DSLR cameras that require user control, our method is invisible to the user. The defocused image and the focused image can be used to perform image demoiréing.

## 6 Conclusion

We have proposed a self-adaptive learning method for moiré pattern removal. Our network predicts a moiré-free clear image from a focused image with moiré patterns, with the help of a corresponding defocused moiré-free blur image, It substantially outperforms state-of-the-art demoiréing methods and joint filtering methods. The moiré-free blur image is easy to obtain through software or hardware. In addition, we have built the first dataset with pairs of focused moiré images and defocused moiré-free images. The future work includes finding more accurate blur kernel estimation and more efficient restoration.

## Broader Impact

Our method improves the quality of photographs taken from a digital camera, by removing moiré patterns to restore an underlying clean, moiré-free image. By design, the algorithm produces restored images that are more faithful to the true scene. This makes the photograph's information more apparent and representative. It is envisioned better quality images will have a positive societal benefit, making visually recorded information more detailed, informative, and useful.

With any approach that improves image quality also comes the risk of negative uses, such as privacy issues. For images of natural scenes, further downstream applications such as surveillance and tracking may become more effective particularly in high-frequency regions of an image where moiré patterns are more likely.

Another consideration is that by removing moiré patterns from pictures taken of digital displays, it may become more difficult to determine, from the image alone, if it is taken of natural scene or of a display such as a computer screen. Potentially this could make it easier for one to take a photograph of a digital screen and claim that the photo is an authentic capture of real scene. However, there may be other indicators if the photo is taken of the screen, particularly if the layout of the LCD elements are visible. Potential future research could explore the difficulty of classifying screen and natural images, with and without demoiréd results produced by the proposed method.

We note, although our method substantially improves the state-of-the-art, it is not perfect and its failures may result in moiré patterns to remain in an image, or be replaced with blurry outputs. As the method is self-adaptive, learning at test-time from two images, we believe the implications of learning from biased data to be minimal.

## Funding Disclosure

The work was supported by the Computer Vision Research Project of Huawei Noah's Ark Lab.

## Footnotes

[2]In this paper, the terms 'moiré image' and 'moiré-free image' denote an image with and without moiré patterns. In addition, 'screen moiré image' means an image whose moiré patterns is caused by digital screen, and 'texture moiré image' denotes an image with moiré patterns caused by high-frequency textures.

[3]The detail processes of the joint optimization and the alternating optimization and their difference are explained in the supplementary material.

[4]The detailed information about the camera models and the screens is described in the supplementary material.

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
