[Supplementary Material]

# Self-Adaptively Learning to Demoiré from Focused and Defocused Image Pairs

## Supplementary Material

**Lin Liu**[1,2]   **Shanxin Yuan**[2*]   **Jianzhuang Liu**[2]

**Liping Bao**[1]   **Gregory Slabaugh**[2]   **Qi Tian**[3]

[1]University of Science and Technology of China
[2]Noah's Ark Lab, Huawei Technologies
[3]Huawei Cloud BU

## 1   Cameras and Screens

We use three cameras and three screens to capture our dataset; please see Table 1 for the specifications. Note that the HONOR Intelligence Screen is a screen with 4K resolution and is used to display images for the *RealTextureMoire* subset. A high-resolution screen helps avoid screen moiré patterns when acquiring texture moiré images.

Table 1: Camera specifications and screen specifications.

| Capture device | | | Display device | | |
|---|---|---|---|---|---|
| Manufacturer | Model | Image Resolution | Manufacturer | Model | Resolution |
| OPPO | R9 | $4608 \times 3456$ | SAMSUNG | S22F350H | $1920 \times 1080$ |
| HONOR | 9 | $3264 \times 1632$ | HONOR | Intelligence Screen | 4K |
| HUAWEI | P30 PRO | $3648 \times 2736$ | HP | E243 | $1920 \times 1200$ |

## 2   The Alternating Optimization Method

The following algorithms (Procedure 1 and Procedure 2) show the joint optimization method and the baseline alternating optimization method compared in the ablation study in Section 5.1 of the main paper. The difference between the joint optimization used for our FDNet and the alternating optimization is shown on the lines 6–10 of Procedure 2, where $D^i$ and $G_k^i$ are updated in an alternating fashion.

---

[*]Corresponding author

---

**Procedure 1** The joint optimization algorithm.

---

**Input:** Focused image $M$ with moiré patterns and defocused blur image $B$ without moiré patterns;
**Output:** Estimated moiré-free image $C$;
 1: Initialize $G_k^0$ and $D^0$ with Gaussian random weights;
 2: Sample $\mathbf{z}$ from the uniform distribution [0,1];
 3: **for** $i = 1$ to $N$ **do**
 4:     $\hat{C} = D^{i-1}(M)$; $\hat{k} = G_k^{i-1}(\mathbf{z})$; $\hat{B} = \hat{C} \otimes \hat{k}$ ;
 5:     $Loss = MSE(\hat{B}, B)$;
 6:     Update $D^i$ and $G_k^i$ simultaneously using the ADAM algorithm;
 7: **end for**
 8: **return** $C = D^N(M)$.

---

---

**Procedure 2** The alternating optimization algorithm.

---

**Input:** Focused image $M$ with moiré patterns and defocused blur image $B$ without moiré patterns;
**Output:** Estimated moiré-free image $C$;
 1: Initialize $G_k^0$ and $D^0$ with Gaussian random weights;
 2: Sample $\mathbf{z}$ from the uniform distribution [0,1];
 3: **for** $i = 1$ to $N$ **do**
 4:     $\hat{C} = D^{i-1}(M)$; $\hat{k} = G_k^{i-1}(\mathbf{z})$; $\hat{B} = \hat{C} \otimes \hat{k}$ ;
 5:     $Loss = MSE(\hat{B}, B)$;
 6:     **if** $i$ is even **then**
 7:        Update $D^i$ using the ADAM algorithm; $G_k^i = G_k^{i-1}$;
 8:     **else**
 9:        Update $G_k^i$ using the ADAM algorithm; $D^i = D^{i-1}$;
10:     **end if**
11: **end for**
12: **return** $C = D^N(M)$.

---

# 3 Results from Real Natural Scenes

We also test our model on a smartphone HUAWEI P30 PRO. We collect some focused and defocused image pairs from natural scenes, where the focused images have texture moiré patterns, as shown in Figure 1. To test on the real world examples, we do some preprocessing, e.g., alignment. We keep the areas where the moire is produced at the same depth. FDNet generalizes well to images taken from natural scenes (not from screens), as the results are moiré-free and the details are retained from the focused moiré image.

Focused image      Defocused image      Demoireing by FDNet

Figure 1: Examples captured from natural scenes.

# 4 Efficiency Comparison among our Model, DIP and Double-DIP

We evaluate the efficiencies of the FDNet and the deep-image-prior methods (DIP and DoubleDIP) on an NVIDIA RTX 2080Ti GPU. The number of iterations for DIP to find an optimal result varies from image to image, and needs to be manually adjusted. In the demoiréing task, DIP takes 1000 iterations. Double-DIP also requires 1000 iterations to converge. Our model does not have the problem of getting worse results when iterations are over some threshold, due to the constraint by the blur image. FDNet converges in about 500 iterations and then its PSNR slightly increases with the iteration number increasing (see Figure 2 for one example). As shown in Table 2, our FDNet has a faster runtime.

| Algorithm | DoubleDIP | DIP | FDNet |
|---|---|---|---|
| Time (s) | 280 | 43 | 30 |
| Parameters (MB) | 3.08 | 2.22 | 2.64 |

Table 2: Efficiency comparison.

# 5 Visualization of Intermediate Results and Blur Kernels

We visualize some intermediate results (see Figure 2), which show that as the number of iterations increases, the moiré patterns gradually disappear. Figure 3 shows the the learnt blur kernels for different image pairs. Note that the learnt blur kernels are learned from scratch and adaptive to each image pair.

| 3: PSNR=15.27 | 5: PSNR=19.50 | 10: PSNR=23.66 | 15: PSNR=24.83 |

| 50: PSNR=28.27 | 100: PSNR=29.36 | 500: PSNR=30.19 | 3000: PSNR=30.72 |

Figure 2: Intermediate results of one example in *SynScreenMoire*. The numbers to the left of the PSNR are the numbers of iterations.

Figure 3: Visualization of estimated blur kernels.

# 6  Examples of our New Dataset

Figures 4 and 5 show some examples of the *RealScreenMoire* subset and the *RealTextureMoire* subset, respectively. Note that the focused images have more details overlaid with moiré patterns, while the corresponding defocused images have no moiré patterns but appear blurry.

Figure 4: Examples of *RealScreenMoire*.

Figure 5: Examples of *RealTextureMoire*.

# 7 Additional Visual Comparisons on *SynScreenMoire*, *SynTextureMoire*, *RealScreenMoire* and *RealTextureMoire*

Figure 6, Figure 7, Figure 8 and Figure 9 show more results on *SynScreenMoire*, *SynTextureMoire*, *RealScreenMoire*, and *RealScreenMoire*, respectively. The main paper presents an analysis of the results for Figures 6 and 7 on *SynScreenMoire* and *SynTextureMoire*.

As shown in Figures 8 and 9, the deep-learning based methods (SVLRM, MopNet and DJF) produce some artifacts near the edges. DJF also tends to over-sharpen the images and exhibits ringing artifacts. The results of DIP have obvious moiré artifacts left, and DoubleDIP has a global color shift from the original input. In addition, the joint filtering methods (GF and MSJF) tend to smooth the high-frequency regions. Our FDNet outperforms all of them.

Figure 6: Visual comparisons on *SynScreenMoire*.

Figure 7: Visual comparisons on *SynTextureMoire*.

Figure 8: Visual comparisons on *RealScreenMoire* (without ground truth).

Figure 9: Visual comparisons on *RealTextureMoire* (without ground truth).