[Reviews · NeurIPS 2020]

Review 1

Summary and Contributions: This paper presents a method for learning to remoire moire effects from only a pair of corresponding moire and blur images. As in previous work on deep internal learning/Deep Image Prior, no other training data is needed: the input pair is used to jointly train image generator and blur generator networks. Despite no access to ground truth supervision and no prior training, this method outperforms previous demoire work. In addition, new datasets of both screen moire and texture moire effects are presented.

Strengths: The combination of a moire input (with mid frequency aliasing but high frequency detail) and a blurry input (with correct mid frequencies but no high frequency detail) is quite clever, given that DIP/Double-DIP alone cannot remove this mid frequency effect. The results are impressive and match or exceed prior work that required training on a large dataset and used ground truth images as supervision. I appreciate the authors’ efforts to capture a dataset focused on texture moire in particular, since this seems to be under addressed in prior work but is probably more relevant to consumer photography in practice. The ablation study in Sec 5.1 of the main paper is valuable and addresses many of the natural questions that come up (what happens if you don’t have the moire input, what happens if you don’t use a generator for the blur kernel, what happens with a simpler model for getting the output from the moire image, etc). Section 5 in the supplement is also useful for understanding the method (I would also like to see the corresponding blur kernel over time as part of Fig 2).

Weaknesses: My main criticism is that this paper could include more signal processing analysis of *why* this correctly trains a network, since this is a subtle point. The interesting aspect of moire (and other aliasing artifacts) is that it induces *mid* frequency errors in the undersampled output signal. Prefiltering with a low pass filter gets rid of it (hence the blur images used here) but postfiltering with a low pass filter cannot, hence the failure of DIP to remove the effect (since DIP relies on the spectral bias of the CNN to learn lower frequencies first). Of course, blind deblurring of the blur input alone without the moire image would also not work since it would most likely converge to an identity blur kernel and blurry output. As I said in “Strengths,” I think this is quite a clever observation. My best guess at what’s happening here is that, given the moire image as input, *removing* the moire effect is fairly low frequency and provides a good local minimum for the networks to converge to. However, there is no comment or math in the paper to make this explicit, which I think is the biggest missed opportunity here and could have added more value in helping explain why this combination of DIP and deblurring recovers a good output image. Smaller comments: Only a few real world cellphone examples are presented in the supplement and they look much less convincing than the real world dataset examples. Real world seems to raise further issues, like the fact that defocus blur will now be spatially varying and the blur/moire images are unlikely to be well aligned when captured by hand. Any image restoration paper should always include corresponding metrics for the input images, since improvement is relative, not absolute. None of the tables here showed PSNR/SSIM/NIQE/BRISQUE of the input moire or blur images. It is implied by the relative performance on the Screen and Texture datasets that texture demoire is a harder task. Is that supported by the raw metrics on the input images being worse? What happened in the “K” ablation? It would be helpful to include an example of how this fails in the supplement, and it might shed additional light on why the method works (does the learned blur kernel need to be biased towards being somewhat smooth?). Speed is a tradeoff with any DIP-like method, here there is a big speedup over naive DIP (which doesn’t work for demoire anyway) but 30s is still a lot slower than a feed-forward network would be. On the other hand, there is no multi-day training phase and no dataset required. I think section 3.3 could probably be omitted without any loss, since it basically just describes the simple gradient descent loop that any reader would expect after reading sections 3.1 and 3.2 and looking at figure 1. At the very least it could be moved to the supplement.

Correctness: As far as I can tell, everything is correct.

Clarity: The paper is clear and flows well (except the strangeness of including section 3.3).

Relation to Prior Work: The related work is well written and extensive, bringing together a variety of related topics (demoireing, deblurring, jointly filtering multiple inputs, internal test-time learning). As far as I know, this is the first work to use moire/blur images together for deep demoireing.

Reproducibility: Yes

Additional Feedback: I said “yes” for reproducibility, but there are some details missing that would prevent exact replication. Maybe I missed it, but what kernel width K is used in practice? The precise architecture is also missing. However, I think that this method is general enough that reimplementing from scratch using only the paper description would probably give good results, even if the U-Net architecture wasn’t exactly the same. Of course, this would also require access to the newly collected datasets. ============== Post-rebuttal feedback: I appreciate the details provided in the rebuttal. However, this is counterbalanced by some of R4's comments, the lack of underlying analysis, and the fact that real world application is limited since real scenes will not have a constant blur kernel due to varying depth. In the end, I maintain my rating.


Review 2

Summary and Contributions: This paper presents a self-learning approach, leveraging deep images priors, to create a moire free image from a sharp moire image and a blurry free moire image. It takes advantage of the fact that moire only occurs in shape images, so a normally shot blurry image will not show the artifact. They show how to use a deep image prior to solve for a blur kernel with a UNET to demoire the image. The results are quite good.

Strengths: This paper presents a clever approach that takes advantage of a particular image pair like previous joint filter approaches. It uses two images that are practical to get and the method works well. I think this is a nice use of the deep prior and overall I like this paper.

Weaknesses: The main weakness is that test-time training is always needed and what is learned can't transfer to other examples. This is an inherent problem of using a deep image prior and on that translates here. Not only is nothing learned in a generalizable way, the approach is timing consuming as about 3 minutes of training are needed for any image pair to be processed.

Correctness: The paper appears to be technically correct

Clarity: The paper is well written and easy to follow. The figures are nice

Relation to Prior Work: Previous work is discussed and compared too. This is done well.

Reproducibility: Yes

Additional Feedback: In previous work, it has been show that it is hard to know how long to train a deep priors for, as some point they can overfit too much and produce a less than optimal results. How did you decide the number of iterations/stopping criteria? How does the network architecture affect the results? Deep image prior works have shown results with architectures other than UNETs. Were others tried? What are the limitations of this approach?


Review 3

Summary and Contributions: This paper presents a solution for "demoireing" an image --- taking an image with moire artifacts, and then producing an image without them. The approach taken in this paper is very different from prior work, as instead of taking a single image as input, the technique takes two images as input: one image that is "in focus" and therefore has moire artifacts, and a second image (perfectly aligned to the first) that is out of focus and so does not have moire artifacts. The paper then presents a deep image prior-inspired system that recovers a moire-free image from those two input images, by optimizing over an encoder-decoder on image 1 and a noise-decoder for a blur kernel such that the first image, when blurred, reconstructs the second image.

Strengths: I found the structure of the proposed architecture very interesting, and potentially useful. The demonstration that using a DIP-like framework to parameterize the blur kernel outperforms the naive baseline of directly optimizing over a learnable kernel (ablation “K”) is genuinely interesting and somewhat surprising. The idea of solving a model recovery problem such as this one by training on a pair of images and using a blur kernel to tie the two images together is thought-provoking. The paper is generally well written, and the evaluation seems solid (with one significant caveat).

Weaknesses: I have many concerns about this paper: 1) The problem statement does not make sense to me, 2) The paper significantly overstates the prevalence of moire in real images, 3) the evaluation against all baselines is fundamentally flawed, as the proposed method takes as input two images and the baselines take only one image. 1) The premise of this paper is that someone is going to take a picture with a camera that is in-focus but has moire, and then take a second picture that is out of focus, and then fuse those two images together to produce an in-focus moire-free image. This does not make sense to me. Moire occurs when the point spread function of a camera is smaller than a pixel on the sensor, which will cause aliasing. When you change the focal length of the camera, you make the PSF larger. I understand that the second image in this pair of images has a very large PSF, so large that it is not in-focus. But if you are already willing to take two images with different focuses, why not simply set the focal length of the camera so that the PSF is slightly larger than a pixel? This is will produce an image that is sharp, but does not have moire. Indeed, this is how virtually all cameras in the world take pictures --- they construct the optics of the camera such that the PSF is wider than a pixel, and they take a single in-focus picture. I do not understand why someone would intentionally design a camera with aliasing and take pairs of pictures with it, when they could simply take a single picture from that camera with the optimal focal length (or just design a normal camera that does not alias, as people do). 2) This paper makes it sound like moire is a significant problem in modern cameras, but this is not the case. We can quickly verify that moire artifacts are uncommon in natural images by looking at how the data used in this paper was generated: they are all photographs of computer screens (a grid of pixels). Virtually all cameras are designed so as to have optical properties that do not cause aliasing to occur on normal natural images. This is accomplished either by having a somewhat "soft" lens, or by having an optical low pass filter on the sensor. It is possible that a solution to this problem might be of interest to a computational imaging conference, where esoteric cameras that intentionally alias their images are explored, but I don't thing the NeurIPS community needs to be made aware of this problem (or furthermore, the two-image variant of this problem that is explored here). 3) If I understand the evaluation of this proposed method correctly, it significantly outperforms prior work. But it seems that all prior works take as input *just a single image*, and the input to this model is two images. This seems to make the comparison not very meaningful --- I'd expect a system that has two images as input to outperform all systems with one image as input, especially since the second image does not contain moire! This ties into my concern #1, as it seems that if the focal length of the second image is set correctly, it is a perfect solution to the problem being posed here, as it will be as sharp as possible while not aliased.

Correctness: The paper contains a number of claims about the nature of moire that I do not believe are correct: Line 1: “Moiré artifacts are common in digital photography” --- This is largely untrue, a photographer is unlikely to ever encounter moire in a photograph unless they are doing something unusual like taking a picture of a screen --- either a computer monitor/tv, or a metal screen door. Virtual all consumer cameras are designed such that the point spread function of the lens and sensor is wider than a pixel, and this optical prefiltering minimizes aliasing. Line 39-40: “...low-pass filters require special hardware design and thus cannot be widely used on smartphones.“ This is incorrect. Optical low pass filters (OLPFs) are very common in consumer camera design, and historically almost all consumer cameras have included them. They are sometimes omitted in modern cameras (often marketed as a way to increase the effective resolution of a camera) but only because the lens of a camera generally serves the same optical prefiltering role. But it is certainly not the case that OLPFs are infeasible to build --- camera manufacturers can build cameras with OLPFs, and have for decades.

Clarity: Besides my previously stated issues, the paper is sufficiently well written.

Relation to Prior Work: Yes

Reproducibility: Yes

Additional Feedback: POST REBUTTAL: After looking at the other reviews and the rebuttal, I stand by 2 or my 3 points in the "weaknesses" section of my review: 1) The problem statement does not make sense to me, 2) The paper significantly overstates the prevalence of moire in real images (my 3rd concern that the evaluation didn't include baselines that use multiple images as input was apparently wrong, or so the rebuttal claims, which I'm willing to believe). The rebuttal says that "Although setting the PSF slightly larger than a pixel can prevent moire patterns, doing so will generate blurry images." This isn't true as far as I know. It's pretty standard to construct a camera where the image is prefiltered to such a degree that moire doesn't occur, but excessive blur isn't introduced. The rebuttal (correctly) asserts that many cameras don't have optical low pass filters, but does not address the core thrust of my argument, which is that those same cameras are designed to not alias by having the low pass prefiltering be part of the optics of the camera. The authors claim in the rebuttal that they were able to produce images that contain moire by shooting actual photographs of something other than a screen with actual cameras (unlike all other images presented in the paper which are either photos of screens, or internet images that appear to be photos of screens) but they chose not to include any images in their rebuttal, which leads me to question this claim.

[Author Response · NeurIPS 2020]

We thank the reviewers for the constructive and extremely helpful comments. We have updated the manuscript and believe it has substantially been improved. We appreciate that all reviewers have recognized our method's novelties by saying it is 'quite clever' (R1), 'a clever approach' (R2), 'very interesting, and potentially useful' (R4), and our effort to build new datasets (R1).

**R1: Signal processing analysis.** Thanks for your comment. We have described it in more detail in our new version. As explained in [11], moire patterns and noise are different; the former are prevalent more in low and mid-frequencies. DIP relies on the spectral bias of the CNN to learn lower frequencies first. So before DIP learns the high-frequency details of the image, moire patterns have appeared in the results of DIP. Mathematically, adding the moire image provides a good local minimum for the network to converge to. In fact, although the moire image is corrupted by moire patterns, it still retains many high-frequency details. The UNET can retain these high-frequency details in the iteration for the reconstruction of a clean image from the blur image that lacks high-frequency details.

**R1: Real world examples.** We did encounter the problems you mentioned in practice. To test on the real world examples, we did some preprocessing, e.g., alignment. The problems have been addressed in some papers, e.g., multi-focus image fusion methods [Pixel Convolutional Neural Network for Multi-Focus Image Fusion. Information Sciences, 2017; Multi-focus Image Fusion with a Deep Convolutional Neural Network. IJLEO, 2018] for image alignment, and depth estimation from focus methods [Depth estimation from focus and disparity. ICIP, 2016] to extract the moire area with the same depth. These methods can be applied in practice to put our model into cameras. However, our paper focuses on how to use focused and defocused image pairs to remove moire patterns.

**R1: Metrics for the input images.** The averages of PSNR/SSIM for the moire input in *SynScreenMoire* and *SynTextureMoire* are 23.08/0.809 and 22.73/0.784, respectively, which are much worse than the demoireing results by our method. Usually, screen demoireing is easier than texture demoireing because screen moire patterns appear throughout the image, whereas in texture demoireing, the moire patterns only appear in high-frequency areas which must be both identified and demoired.

**R1: The 'K' ablation, and Section 3.3.** A failure example has been shown in the 'K' of Fig. 3. Adding a fully-connected network to learn the blur kernel can make the image satisfy the total variation prior. Therefore, the full model can smooth the noise without damaging the original image content. We have updated Section 3.3 accordingly.

**R1: Speed.** Our method does not require multi-day training and large training data. But similar to other DIP-like methods, ours is slower than feed-forward networks. We will address this issue in our future work.

**R2: Number of iterations.** Our model does not have the problem of getting worse results when iterations are over some threshold, due to the constraint by the blur image. In all our experiments, the number of iterations is set to 3000 empirically.

**R2: Other architectures.** We also tried the encoder-decoder and ResNet. Their PSNRs are 0.11dB and 1.05dB, respectively lower than UNET. Many other image restoration methods have shown that UNET has advantages over the two structures.

**R2: Limitation.** The main limitation of the work is that the test-time training is slower than pre-trained networks.

**R4: Premise, PSF.** Although setting the PSF slightly larger than a pixel can prevent moire patterns, doing so will generate blurry images. We also note in practice, the PSF is a spatially variant function, and additionally depends on wavelength, so it not possible to engineer an ideal PSF that produces a uniformly sharp image without moire patterns. In contrast, our algorithm can produce clear and moire-free images.

**R4: Significance in modern cameras.** In the professional photography community, the use of optical low pass filters (OLPFs) as suggested by the reviewer is in fact controversial. Although OLPFs do greatly reduce moire patterns, they come at the expense of image sharpness, and increasingly manufacturers remove OLPFs to improve acutance. Mainstream companies (Nikon, Canon, Pentax, Sony) now all offer multiple DSLR cameras without OLPFs, e.g. Nikon D5600, D7500, D500. Most mainstream smartphone manufacturers (Apple, Google, Huawei) use Sony image sensors (e.g. IMX586, IMX363, etc.) which also do not include OLPFs instead relying on software to reduce moire effect. We confirmed this by producing moire patterns by photographing high frequency patterns using flagship mobile phones (Google Pixel 4, Huawei P40, OPPO Reno3). Further, we note the presence of moire artifacts is often used to justify lowered smartphone camera ratings as part of DXOMark reviews, further confirming the commercial significance of this problem. Please note, the SynTextureMoire dataset does not rely on photographing a screen and focuses on moire in high frequency textures.

**R4: Evaluation: single vs two images.** Please note, we *do* compare to other methods that also take two images as input, namely GF, DJF, SVLRM, and MSJF; please revisit Table 3 and Figure 4. We also compare to recent single image demoireing methods for comparisons to the SOTA. As this is the first paper to use both moire/blur images to demoire, there are no demoire-specific methods to compare to that accept these two inputs. However, based on the reviewer concern, we conducted a 'fair' experiment by feeding both the blur image and the moire image to the previous single image demoireing models for training and testing, and found doing so helps a little but still underperforms our proposed method. This result has been added to the revision.

**R4: Deployment of our method.** In Section 5.3, we describe how to easily apply our method to smartphones. In the typical capture mode, the camera applies an auto-focus algorithm, which can be modified to save an additional defocused image during the focusing process. In practice, it is not necessary to take an image with moire patterns and then take a defocused image.

**R4: Relevance to NeurIPS.** We note the list of NeurIPS 2020 Subject Areas includes "Computational Photography", and further our work is related to unsupervised learning. Given the prevalence of mainstream DSLR and smartphone cameras without OLPFs, the moire problem is both ubiquitous and pressing. We believe our solution is of interest to the NeurIPS community.

**R4: Line 39-40.** The low-pass filters mentioned in this paragraph are variable low-pass filters (line 38), where the word "variable" is missing. We have added it in the revision. Thank you for pointing this out.

[Meta-Review · NeurIPS 2020]

Two of the reviewers recommended accepting this paper and a third recommended rejection. The two main concerns of the negative reviewer were the experimental comparisons and the importance of Moire patterns. The authors convincingly addressed the first point in their rebuttal, and in the discussion the reviewer acknowledged this fact, yet stood by the assertion that Moire patterns are not of major importance in modern cameras. I tend to agree with the other two reviewers: while Moire patterns are somewhat of a niche phenomena, they are still interesting and the approach here is novel and interesting. So I am happy to recommend accepting the paper but urge the authors to do a better job of motivating the problem to the general NeurIPS audience. Some of the material in the rebuttal can be useful to add to the revised paper in this respect.